# Urethral Sheath to Evacuate Blood Clots through Mitrofanoff Appendicovesicostomy

**Marcello Della Corte** [1,2,*], **Erica Clemente** [3], **Mattia Sibona** [4], **Elisa Cerchia** [2], **Berenice Tulelli** [5], **Paolo Gontero** [4] **and Simona Gerocarni Nappo** [2]

1. Division of Urology, Department of Oncology, School of Medicine, San Luigi Gonzaga Hospital, University of Turin, Regione Gonzole 10, 10043 Orbassano, Italy
2. Division of Pediatric Urology, Regina Margherita Hospital, 10126 Turin, Italy
3. Department of Medical Sciences, University of Turin, Via Verdi 8, 10124 Turin, Italy
4. Department of Urology, AOU Città Della Salute e Della Scienza di Torino, Molinette Hospital, University of Turin, 10126 Torino, Italy
5. Department of Pediatric Urology, Children and Mother Hospital, 69500 Lyon, France
* Correspondence: dellacortemarcello@gmail.com; Tel.: +39-011-902-6477

**Abstract:** Background: the Mitrofanoff appendicovesicostomy provides a catheterizable submucosal tunnel between umbilicus and bladder (or neobladder). In patients with a closed bladder neck, the Mitrofanoff channel is the only way to access the bladder. We describe our case of a 17 year-old girl with a Mitrofanoff appendicovesicostomy and a previous surgical closure of the bladder neck and who developed a large bladder clot due to hematuria after a surgical cystolithotomy in an augmented bladder; Methods: after an unsuccessful trans-appendicovesicostomy bladder washing, the endoscopic evaluation was performed using a 14 Ch rigid cystoscope and surrounded by its own urethral sheath. The clot was progressively fragmented through the cystoscope under direct vision. Clot fragments were aspirated to obtain a complete evacuation; Results: the urethral sheath prevented damages to the appendicovesicostomy, allowing at the same time repeated accesses of the cystoscope into the neobladder and ensuring the procedural success. The postoperative period was uneventful, and the neobladder catheter was removed after two days. Neither channel stenosis nor anastomosis dehiscence nor incontinence were reported after five months; Conclusions: the use of urethral sheath 14 Ch through an appendicovesicostomy preserves both the stoma and the channel, making possible endoscopic procedures such as blood clot evacuation into the neobladder.

**Keywords:** Mitrofanoff; appendicovesicostomy; clots; hematuria; augmented bladder; urethral sheath; bladder tamponade

## 1. Introduction

The Mitrofanoff appendicovesicostomy (or "Mitrofanoff channel"), first described in 1980 [1], is a urinary continent diversion aimed to treat complex urological conditions [2], such as bladder exstrophy, neurogenic bladder and vesicovaginal fistula. This surgical technique provides a continent catheterizable tunnel between the umbilicus and the bladder (or neobladder) to achieve the goals of continence and painless auto-catheterization [1,3]. During the bladder filling, the progressive increase of intravesical pressure is transmitted to the submucosal tunnel of the appendix, determining its lumen occlusion, and thus ensuring continence [1]. In patients who have undergone a contextual bladder neck closure, the Mitrofanoff channel is the only way to access the bladder (or neobladder) lumen. Therefore, in case of complications, such as the presence of stones or clots, the trans-Mitrofanoff passage of endoscopic instrumentation has enhanced concerns about the possible adverse effects, such as continence mechanisms impairments and iatrogenic stomal stenosis.

A frequent complication in augmented bladders is the formation of bladder stones as the ileal mucus act as a facilitator. The main risk factors for stones formation: persistent

post-catheterization residual urine, recurrent urinary tract infections and presence of foreign bodies. Nowadays, trans-Mitrofanoff endoscopic approaches have been described exclusively for stones management; however, the open approach is recommended as the safest and easiest technique, especially to treat large and multiple calculi [4–6]. In particular, when an access through the Mitrofanoff channel is required, an Amplatz sheath allows the channel protection and the free drainage of the irrigation system, without reaching high intravesical pressures (in particular in augmented bladders) [6].

Bladder clot retention, also known as bladder tamponade, is a rare condition consequent to a massive hematuria. It usually presents with a cohort of typical clinical signs and symptoms: suprapubic discomfort or bladder overdistension, cystospasm, worsening of hematuria and dysuria. Once abdominal ultrasonography (US) has confirmed the diagnosis, manual bladder irrigation with large-bore catheters constitutes a feasible and suggested strategy to progressively clear the urine and allow the aspiration of clots. Computed tomography (CT) should precede any invasive maneuver in a trauma setting to avoid the risk of further damages on an injured lower urinary tract [7]. In neobladders, the symptomatology is usually more nuanced, consisting of abdominal pain or tension and dysuria or difficulties in catheter drainage.

Different techniques have been described to treat bladder tamponade, for example, the suction with large-bore catheters and thoracic drainages or other surgical tools connected with intermittent suction (syringe or Ellik) or continuous vacuum systems. In case of failure, intravesical agents may help to repeat the procedure as they reduce the clots' size, facilitating their aspiration. Nevertheless, in order to prevent the main long-term complications such as stoma stenosis or conduit incontinence, large caliber catheters cannot be used through Mitrofanoff channels; therefore, the suggested highest catheter size is 14 Ch [3]. Endoscopic or percutaneous approaches may be taken into consideration in refractory cases, while open surgery represents the last choice.

The percutaneous approach and open cystotomy are also the eligible strategies in case of non-catheterizable urethra (surgical closure of bladder neck, severe urethral stricture and neobladder), but they are still invasive—even if minimally—approaches; therefore, they require the operatory room availability and the anesthesiologic support [7].

This work aims to describe our challenging big clot evacuation from an augmented ileal bladder though a Mitrofanoff channel. Among the current literature, no similar cases have been described so far.

This case report respects CARE Guidelines, reflecting the point-by-point 2013 CARE Checklist [8]. Patient medical history, perioperative and postoperative clinical data and images are described. Surgical instruments, tools' data and treatment are reported. The patient perspective has been assessed with a qualitative self-evaluation of her personal clinical experience, while pain was objectively assessed with a numeric rating scale (NRS), from 0 ("no pain") to 10 ("worst pain imaginable"). The limits of the described approach are strengthened at the end of the discussion section.

Informed consent for publishing purposes was obtained from the patient.

## 2. Case Report

### 2.1. Case Presentation

A 17-year-old girl, weight 38 kg and height 140 cm, developed hematuria three days after a surgical cystolithotomy for a 30 mm bladder stone. After an unsuccessful bladder washing, she was administered with oral tranexamic acid resulting in the successive end of hematuria. Subsequently, she developed abdominal pain, NRS 7/10, and inadequate bladder drainage at intermittent catheterization, suggestive of bladder tamponade. The bladder lavage allowed a bare evacuation of small clots.

### 2.2. Medical History

The patient was born premature at 34th gestational week, from a planned caesarean section (CS), due to a maternal malignant neoplasm. She presented a congenital anorectal mal-

formation (cloaca) that was immediately treated with colostomy. At two years of age, she underwent anorectoplasty and vaginoplasty. Bladder neck closure occurred as a complication, and a permanent vesicostomy was performed, resulting in total urinary incontinence.

Two years before our evaluation, at the age of 16, the patient was treated with bladder augmentation (ileum enterocystoplasty) together with Mitrofanoff appendicovesicostomy.

During the current admission, she underwent uncomplicated cystolithotomy of a three-centimeter bladder stone.

### 2.3. Diagnostic Assessment

Abdominal US showed a large clot filling the bladder, varying in position depending on the patient decubitus. The clot had a 10 cm diameter as a main projection (Figure 1a,b).

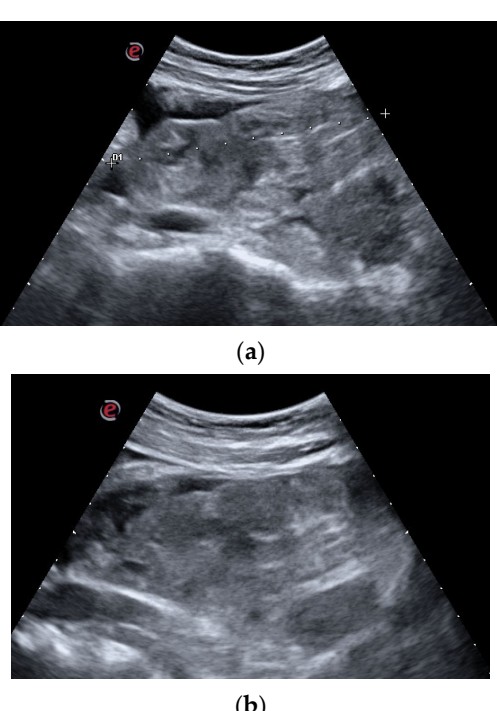

(**a**)

(**b**)

**Figure 1.** (**a**,**b**). Diagnostic Assessment: US images of the large clot occupying the entire bladder lumen. The complex organization and the tight adhesion of its major components are clearly appreciable both in the sagittal scan (**a**) and in the transversal scan (**b**).

Considering the recent open surgery, the persistent abdominal pain and the bladder overdistension, CT scan was performed with the administration of iodinated contrast media (ICM). The late phase showed a big clot occupying the entire bladder lumen, with an ICM accumulation around the clot that appeared hypodense.

### 2.4. Surgical Management

The clot was neither susceptible to a conservative approach, due to its size, nor likely to be managed through a trans-catheter evacuation even after several failed attempts. Large caliber catheters were avoided in order to not damage the Mitrofanoff channel.

The patient was transferred to the operatory room. She underwent general anesthesia, and she was placed in a supine position. The trans-appendicovesicostomy catheter was coaxially pierced with a hydrophilic nitinol 0.035-inch guidewire (Boston Scientific®—Marlborough, 300 Boston Scientific Way, MA, USA).

A 10 Ch rigid cystoscope surrounded by its own ureteral 14 Ch sheath (Karl Storz Se & Co. KG®—78532 Tuttlingen, Germany) was gently inserted through the appendicovesicostomy and coaxially to the guidewire, which was removed after the bladder wall attainment.

The preliminary endoscopic evaluation confirmed the hematic nature of the clot without any sign of active bleeding (Figure 2a,b).

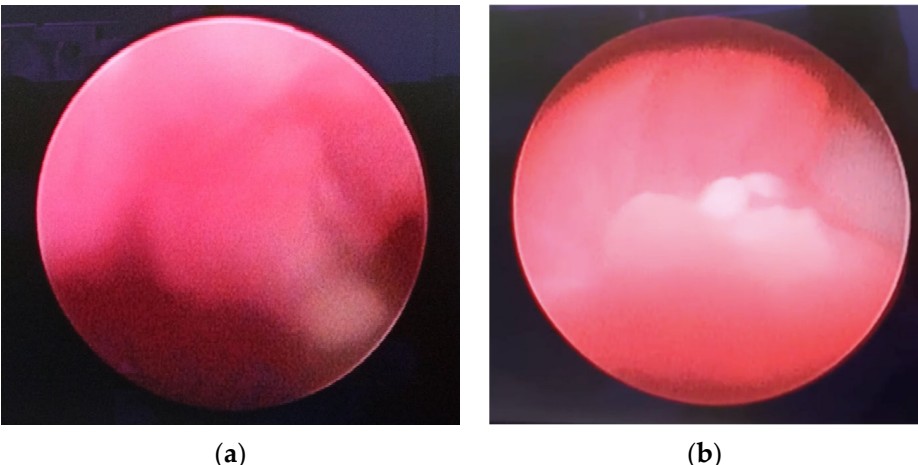

(**a**) (**b**)

**Figure 2.** (**a**,**b**). Endoscopic view: endoscopic appearance of the clot. The light red shade of color suggests the late formation of the clot and, thus, its friability. No signs of active bleeding are visible. The 30° endoscopic optics ensured correct angulation of the view, overcoming the reduction in movement caused by the Mitrofanoff channel insertion angle.

Under direct vision, the clot was progressively fragmented with the aid of the cystoscope, compressing the clot against the neobladder wall and weakening it. The vacuum effect of the 60 mL syringe permitted a progressive clot reduction into smaller fragments, which were progressively evacuated (Figure 3).

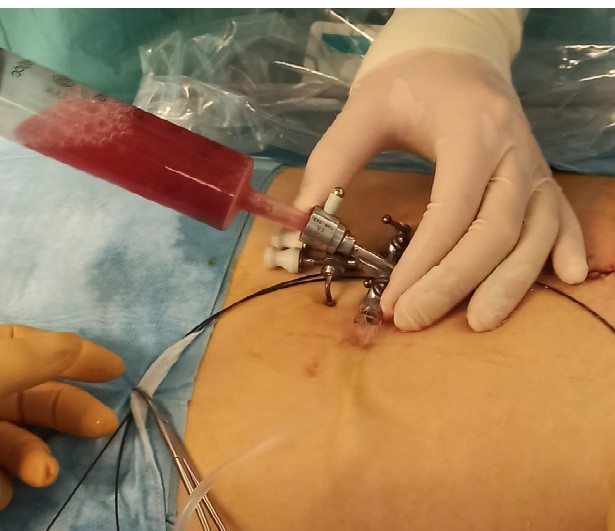

**Figure 3.** Intraoperative image: the evacuative approach is clearly shown. The 14 Ch urethral sheath is inserted into the Mitrofanoff. All operative channels are closed in this phase, while a 60 mL syringe is whipped at the optical access. The surgeon's left hand firmly holds the sheath, while the right one (not visible in this picture) is dedicated to clot aspiration.

Due to the limited degrees of movement, in order to not damage the appendicovesicostomy, the contemporary US examination was performed till complete clot disruption was achieved (Figure 4a–d).

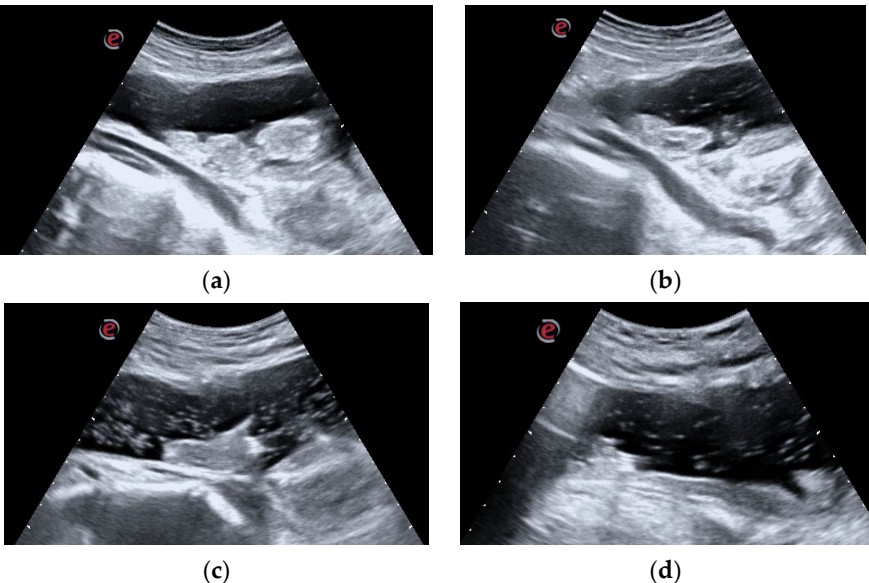

**Figure 4.** (**a**–**d**). Intraoperative US evaluation: the intraoperative US examination improved the assessment of the progressive clot disruption till its complete disappearance. In image (**a**), the main clot components are clearly visible. During the successive phases, images (**b**,**c**), clot dimensions progressively decrease, while a corpuscular suspension occupies the filling liquid. Finally, image (**d**), the clot is no more visible and only small fragments float in the lumen.

The sheath of the cystoscope was removed under direct vision, ensuring the optimal conditions of the Mitrofanoff channel and of the surgical anastomosis.

In the end, a trans-Mitrofanoff 14 Ch catheter was placed to ensure the continuous drainage of the neobladder and to monitor the clarity of the urine.

### 2.5. Results, Outcomes and Follow-up

The procedure was performed by two urologists, and the complete clot removal was achieved in 130 min. The urethral sheath preserved the appendicovesicostomy from the continuous passages of the cystoscope and its swinging movements.

Postoperative period was uneventful. Neobladder catheter was removed after two days. Neither channel stenosis nor anastomosis dehiscence nor incontinence were reported after five months.

### 2.6. Patient Perspective

The patient reported a full satisfaction for the received treatment, and in particular, she appreciated the minimal invasiveness of the procedure and the prompt postoperative recovery. The procedure did significantly reduce the pain perception as, post-operatively, the patient experienced pain up to NRS 3/10. Therefore, no administration of pain therapy was required.

## 3. Discussion

The Mitrofanoff appendicovesicostomy represents a challenge in performing endoscopic procedures in patients with a previous bladder neck closure.

The bladder clots management usually consists of drainage with large-bore catheters and manual irrigation [7,9]. In case of failure, surgical approaches, intravesical agent instillation and endoscopic procedures have been described for bladder clot evacuation, and the preferrable choice depends on the characteristics of the patient, the surgeon's expertise and the hospital equipment availability.

In the presented case, as hematuria was developed due to a recent cystolithotomy, a re-do open surgical approach would not have represented a preferential strategy.

Intravesical agents are a group of substances that, after a trans-catheter bladder instillation, reduce the consistence or disintegrate the blood clots due to their enzymatic or chemical activity [7]. The first attempt in 1993, described by LaFave and Decter, involved a successful clot dissolution after intravesical urokinase in two boys [10]. Korkmaz et al. used streptokinase in a 12-year-old patient, without encountering changes in hemodynamic and coagulation indices [11]. Lastly, Olarte et al. managed a critical neonate in ECMO support with alteplase to dissolve a big clot [12]. Similar approaches with different agents, such as chymotrypsin plus sodium bicarbonate and hydrogen peroxide, have been described in adults and lack data in pediatric patients [7]. In the presented case, intravesical agents instillation was not taken into account as the patient had an ileocystoplasty. These drugs could be absorbed by ileal villi and enter in systemic circulation, carrying the risk of unpredictable effects on blood coagulation [7]. Truthfully, ileal mucosa exposed to urine develops many changes, such as villous atrophy, reducing its absorptive properties. The metabolic ileal bladder activity may also be influenced by other factors (e.g., mucus, neo-urinary tract microbiota [2], enzymes and lymphatic circle absorption); therefore, further studies are required to assess these features [13]. Due to the lack of proper knowledges among neobladder resorptive abilities, an intracavitary treatment was not chosen.

Although the urokinase and streptokinase properties through an oral administration are not significant [14–17], their eventual absorption after a direct ileal administration has never been studied. In particular, this anatomical district does not endure most of the digestive processes encountered after an oral administration. Furthermore, in consideration of the recent cystolithotomy, an intravesical plasminogen activator administration would have provoked an additional bleeding.

Apart from the unexpected systemic effects, other local consequences would have showed up in case of different intravesical agent administrations. For example, intravesical hydrogen peroxide instillation would have released high volumes of gaseous oxygen, which would have been inappropriate for an augmented bladder that had recently undergone cystotomy. The consequent uncontrollable distension of neobladder walls and its stretching effects on the recent stitches are risk factors for bladder rupture [18]. Lastly, the relative safety attributed to intravesical agents cannot be extended to the field of neobladders because of the absence of data about these patients.

Endoscopic approaches are based on Ellik suction and sometimes preceded by the physical disruption of clots using morcellator or resectoscope. When the urethra is not patent, percutaneous endoscopic approaches and open cystotomy represent the eligible alternatives [7]. The use of a resectoscope through the Mitrofanoff channel would have mechanically stressed the appendicostomy because of the required swinging movements. In addition, Ellik suction would have represented a risk factor for neobladder rupture because of the recent cystotomy and the presence of fresh stitches as stated above.

Lastly, an endoscopic transurethral retrograde approach was not feasible because of previous bladder neck closure.

The patient's characteristics (a closed bladder neck, an ileal augmented bladder and a Mitrofanoff appendicovesicostomy) forced us to choose a conservative and minimally invasive endoscopic approach. As described in this case, using a sheath in the appendicovesicostomy allowed the preservation of both stoma and Mitrofanoff conduit, assuring safe clot fragmentation and removal. A similar technique was described for stone fragmentation using Amplatz 18–28 Ch sheath by Thomas et al. [6] However, the cystoscope 14 Ch sheath in this case allowed safe placement in the Mitrofanoff channel, with no risk of conduit damages.

Although this case was managed using general anesthesia to avoid patient's discomfort and to promptly manage any possible complication, considering the recent open surgery, the minimal invasiveness of this procedure would have allowed a safe executability even in an ambulatory settings. In that case, pain relieve techniques, such as nitrous oxide, virtual reality analgesia and other sedation techniques, may be considered [7]. In addition, this

type of approach would be cheaper than a surgical one and does not require an high level of expertise as opposed to the percutaneous, endoscopic or surgical ones.

This is the first described case of a bladder clot managed with an endoscopic approach using an urethral sheath through a Mitrofanoff channel.

*Limits of This Case*

The case management has some limits. Through this work, we offer an objective description of a surgical management. Nevertheless, the main limit is its anecdotical singularity. Any type of high-scale application needs to be thoroughly evaluated and cannot be carried out uncritically. Its uniqueness only offers a new perspective of treatment in some selected cases, but a specific case evaluation is strongly suggested if anybody wishes to extend the proposed technique to other patients. Like most case reports, a generalization is not possible, and the clinical results may be different than the expected ones. Their overinterpretation or misinterpretation may induce an "anecdotal fallacy [19,20]".

## 4. Conclusions

The bladder blood clots evacuation in a closed-neck augmented bladder with a Mitrofanoff channel presents some challenges related to its management. The absence of a natural endoscopically explorable channel impends the use of a resectoscope, and the Mitrofanoff tunnel is highly delicate and therefore suitable to fulfill this scope. The use of urethral sheath simplifies the clots suction and improves the safety of the minimal endoscopic maneuvers. This conservative technique avoids the need of an open surgical approach and does not damage the continence mechanisms of the Mitrofanoff channel.

In the described case, a big blood clot in an augmented bladder, not manageable through a transurethral endoscopy, was evacuated using an urethral sheath placed into the Mitrofanoff channel, without compromising the continence mechanisms. Even if this case was managed under general anesthesia, the minimal invasiveness allows its feasibility in an outpatient setting, reducing the operative room-related costs.

To our knowledge, this is the first reported case of neobladder tamponade managed in this way thus far.

**Author Contributions:** Conceptualization, M.D.C., E.C. (Erica Clemente), M.S. and S.G.N.; methodology, M.D.C., E.C. (Erica Clemente), M.S., E.C. (Elisa Cerchia), B.T., P.G. and S.G.N.; software, M.D.C., E.C. (Erica Clemente) and S.G.N.; validation, S.G.N.; formal analysis, E.C. (Erica Clemente), E.C. (Elisa Cerchia), B.T. and S.G.N.; investigation, M.D.C., E.C. (Erica Clemente), M.S., E.C. (Elisa Cerchia), P.G., P.G. and S.G.N.; resources, M.D.C., E.C. (Erica Clemente), M.S., E.C. (Elisa Cerchia), B.T., P.G. and S.G.N.; data curation, M.D.C., E.C. (Erica Clemente) and S.G.N.; writing—original draft preparation, M.D.C., E.C. (Erica Clemente), M.S., E.C. (Elisa Cerchia), B.T., P.G. and S.G.N.; writing—review and editing, M.D.C., E.C. (Erica Clemente), M.S., E.C. (Elisa Cerchia), B.T., P.G. and S.G.N.; visualization, M.D.C., E.C. (Erica Clemente) and S.G.N.; supervision, S.G.N.; project administration, M.D.C. and S.G.N.; funding acquisition, M.D.C. All authors have read and agreed to the published version of the manuscript.

**Funding:** This research received no external funding.

**Institutional Review Board Statement:** Not applicable.

**Informed Consent Statement:** Written informed consent was obtained from the patient to publish this paper.

**Data Availability Statement:** The data presented in this study are available on request from the corresponding author. The data are not publicly available due to privacy.

**Conflicts of Interest:** The authors declare no conflict of interest.

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
