# Peer review of "Urethral Sheath to Evacuate Blood Clots through Mitrofanoff Appendicovesicostomy"

_2673-4095, doi:10.3390/surgeries4020030_

Round 1

Reviewer 1 Report

Title: Urethral Sheath To Evacuate Blood Clots Through Mitrofanoff  Appendicovesicostomy

The paper can be accepted after these comments as

1) Provide more literature details in the introduction section.

2) Provide the organization of the paper at the end of introduction section

3) What is the novelty of this work? Please highlight it

4) Provide more details of the figures

5) Conclusions are so short. Provide the final results that you have been achieved in conclusion

Author Response

Dear Reviewer,

We sincerely thank you for your accurate evaluation of our manuscript and for offering us these suggestions. 

We deeply evaluated all of them and we offer below a detailed answer analyzing point-by-point your review report.

Q: 1) Provide more literature details in the introduction section.

A: Thanks for your comment. Honestly, we did not completely understand your suggestion. After some repeated readings of the mentioned works and of our manuscript, we believe that all the notions are properly attributed to a right reference, as well as all the references are appropriate for their referring contents. We just added the missing information. In case the current contents do not fit your purposes, we will be grateful if you offer us further details.

Q: 2) Provide the organization of the paper at the end of introduction section

A: Thanks for your suggestion. The organization of the paper was partially defined in “materials and methods” section. According to other suggestions deriving from the peer review process, in order to follow MDPI guidelines and to respect the structure of case reports, we deleted the paragraph title “materials and methods”, then we joined its content to “introduction” section, adding the proper details as you suggested us.

Q: 3) What is the novelty of this work? Please highlight it

A: Thanks for your comment. The novelty of this work was only implicitly mentioned. Now the main concept have been better stressed at the end of “discussion” and “conclusions” sections.

Very useful observation. 

Q: 4) Provide more details of the figures

A: Thanks to your comment we noticed the legends were poor. Now the figures have been more critically equipped of details as suggested. 

Q: 5) Conclusions are so short. Provide the final results that you have been achieved in conclusion

A: Thanks for your comment. Conclusions have been expanded according to this comment and the previous n°3.

Lastly, we thank you again for your helpful report. We hope that you will agree with all of the changes proposed. If not, we remain at your disposal for any further improvement required. 

Best Regards

Reviewer 2 Report

The clot was progressively fragmented through the cystoscope and a urethral catheter under direct vision - rows 23 and 24 may provoke confusion as the patient does have bladder neck closure. Maybe it would be helpful to rephrase this statement.  

Author Response

Dear Reviewer,

We thank you a lot for your deep evaluation of our manuscript.

Sincerely congrats, you were the only one able to intercept a typing typo of the first draft. The sequence <<and a urethral catheter>> was deleted, because it was an incorrect content and it would have altered the correct comprehension of the entire work.

Thank you again.

Best Regards

Reviewer 3 Report

First of all, I suggest that you include the manuscript in the "technical note" category instead of the "case report" category.

Regarding the structure, I suggest you look at one work from the mentioned category ( https://www.mdpi.com/2227-9067/9/5/640 ). If, like the above, you have a video showing your procedure, be sure to add it to the suppl. materials.

Adapt the abstract and the manuscript itself to consist of an introduction, case presentation (+description of the technique), discussion, and conclusion.

The introduction needs to be significantly reduced. Throw out the well-known facts about the Mitrofanoff appendicovesicostomy, and focus on the challenge of getting the blood clot out of the bladder.

Remove the materials and methods section.

Do not divide the case report section into subsections (3.1, 3.2, etc…)

The discussion and conclusion are well structured, but pay attention to the English language and that the same references are not repeated sentence after sentence (eg reference 7), etc.

Moderate corrections are necessary.

Author Response

Dear Reviewer,

We sincerely thank you for your accurate evaluation of our manuscript and for offering us these suggestions.

We deeply evaluated all of them and we offer below a detailed answer analyzing point-by-point your review report. We sincerely apologize if we cannot consider all of them, since some are in contrast with the others offered during the peer review process. Nevertheless, we offer below a detailed description of all the highlighted critical issues plus our corrections and improvements.

Q: First of all, I suggest that you include the manuscript in the "technical note" category instead of the "case report" category.

A: Thank you for your suggestion. The type of work format you indicated may fit our manuscript. Nevertheless, we did not change the type of manuscript, since during the peer review process this type of work was approved from all the involved figures.

Q: Regarding the structure, I suggest you look at one work from the mentioned category ( https://www.mdpi.com/2227-9067/9/5/640 ). If, like the above, you have a video showing your procedure, be sure to add it to the suppl. materials.

A: Thank you for have offered this useful material. We did not add any video, because we did not record the operation. 

Q: Adapt the abstract and the manuscript itself to consist of an introduction, case presentation (+description of the technique), discussion, and conclusion.

A: Thanks for these suggestions. For the same reasons above, we did not change the structure of the abstract, since it respects MDPI Guidelines for case reports (https://www.mdpi.com/about/article_types) and reflects the structure of the offered template (https://www.mdpi.com/journal/surgeries/instructions).

Q: The introduction needs to be significantly reduced. Throw out the well-known facts about the Mitrofanoff appendicovesicostomy, and focus on the challenge of getting the blood clot out of the bladder.

A: Thank you for your suggestion. Unfortunately this is in contrast with the comment of another reviewer. Therefore, we expect for an Academic Editor's overview and decision on this aspect.

Q: Remove the materials and methods section.

A: Thank you for the suggestion. The section has been removed, but its content did not totally disappear, according to another reviewer's report. 

Q: Do not divide the case report section into subsections (3.1, 3.2, etc…)

A: Thank you for your comment. We prefer to maintain the current structure in subsections. We believe that the division into subsections offers a more schematic and less boring reading and can be useful in a didactic context, since it reflects the CARE Guidelines Checklist.

We hope you could agree with us.  

Q: The discussion and conclusion are well structured, but pay attention to the English language and that the same references are not repeated sentence after sentence (eg reference 7), etc.

A: Thank you for the evaluation. Brief additions were made to discussion and conclusion sections according to the other reviewer suggestions and we corrected the repeated references. English editing has been carried out.

Lastly, we thank you again for your helpful report. We hope that you will agree with all of the changes proposed. If not, we remain at your disposal for any further improvement required.

We hope that you would justify the missing changes since we found very arduous to make all the reviewers suggestions match. 

Best Regards

Round 2

Reviewer 1 Report

The paper can be accepted in present form